# Pulmonary Health Effects of Indoor Volatile Organic Compounds—A Meta-Analysis

**DOI:** 10.3390/ijerph18041578

**Published:** 2021-02-07

**Authors:** Kyle L. Alford, Naresh Kumar

**Affiliations:** 1Department of Public Health Sciences, University of Miami, Miami, FL 33136, USA; kla86@miami.edu; 2Department of Public Health Sciences, Division of Environmental Health, Miller School of Medicine, University of Miami, Miami, FL 33136, USA

**Keywords:** VOCs, indoor air pollution, pulmonary disease, asthma, wheezing

## Abstract

Volatile organic compounds (VOCs) are commonly found in consumer products, including furniture, sealants and paints. Thus, indoor VOCs have become a public health concern, especially in high-income countries (HICs), where people spend most of their time indoors, and indoor and outdoor air exchange is minimal due to a lack of ventilation. VOCs produce high levels of reaction with the airway epithelium and mucosa membrane and is linked with pulmonary diseases. This paper takes a stock of the literature to assess the strength of association (measured by effect size) between VOCs and pulmonary diseases with the focus on asthma and its related symptoms by conducting a meta-analysis. The literature was searched using the PubMed database. A total of 49 studies that measured VOCs or VOC types and pulmonary health outcomes were included in the analysis. The results of these studies were tabulated, and standard effect size of each study was computed. Most studies were conducted in high-income countries, including France (*n* = 7), Japan (*n* = 7) and the United States (*n* = 6). Our analysis suggests that VOCs have a medium-sized effect on pulmonary diseases, including the onset of asthma (effect size (or Cohen’s d) ~0.37; 95% confidence interval (CI) = 0.25–0.49; *n* = 23) and wheezing (effective size ~0.26; 95% CI = 0.10–0.42; *n* = 10). The effect size also varied by country, age and disease type. Multiple stakeholders must be engaged in strategies to mitigate and manage VOC exposure and its associated pulmonary disease burden.

## 1. Introduction

Chronic pulmonary diseases, including the onset and persistence of asthma and chronic obstructive pulmonary diseases (COPD), and their associated morbidity and healthcare costs are on the rise. Moreover, these diseases are associated with reduced productivity and mortality worldwide [1]. In 2019 alone, 339 million new cases of asthma were reported worldwide, of which 417,918 resulted in death [1]. Common symptoms associated with asthma are coughing (with or without phlegm), bronchial inflammation and shortness of breath, and the symptom severity varies from person to person [2]. There are multiple social and economic consequences of asthma morbidity, such as adverse effects on educational performance by preventing school attendance and negative economic productivity due to loss of work [3]. Studies show that asthmatic children miss anywhere between 2.1 and 14.8 days of school depending on the severity of symptoms and socioeconomic status. This is because affluent asthmatic children are less likely to miss school than their less affluent counterparts [3]. There are other costs associated with asthma, including healthcare costs. There was an average cost of USD 1502 (95% CI: USD 1493–1511) for an asthma-related emergency room (ER) visit in 2008, and the total national expenditure on asthma-related ER visits was USD 1.59 billion in 2009 [4].

The extant literature suggests that both genetic and environmental factors play roles in the onset, progression, persistence and severity of asthma [5]. Several studies show that indoor air pollution (IAP), including particulate matter of varying sizes, volatile organic compounds (VOCs), smoke, and allergens (mold spores and endotoxins), are associated with the persistence and severity of asthma morbidity [1,2,5]. Low-income countries (LICs) have different risk factors than high-income countries (HICs), as sources and types of IAP vary across these countries [6]. For the purposes of this study, HICs are defined as countries within the highest twenty nominal GDPs, and LICs are defined as countries within the lowest twenty nominal GDPs [7]. The distinction between HICs and LICs is imperative, as HICs spend more on household furnishings [8,9]. These furnishings can include, but are not limited to, paints, glues, resins and floor polishes. All of these activities and products emit harmful VOCs [10]. HIC households are also typically sealed and rely on central air conditioning (AC), which circulates the same air indoors [11]. Thus, the concentration of indoor pollutants can build up in the absence of effective ventilation and/or filtration if new sources of VOCs are introduced, such as the addition or installation of new furniture, painting, carpeting etc. [11] A study shows that people in HICs spend only 2% of their time outdoors, which can result in their chronic exposure to indoor air pollutants, including VOCs slowly being released from common consumer products and furniture [12].

The inhalation of VOCs is associated with different adverse health effects. VOCs, such as propylene glycol and glycol ethers (PG), benzene and formaldehyde, have high levels of reactivity with the epithelial lining of the respiratory tract and mucous membrane [10]. The mucous membrane is composed of mucin, a polymer of glycoproteins, which are rich in amine groups [10,13]. These amine-rich regions are nucleophilic sites which react with the C–O polar bonds that are commonly found in VOCs [13]. A continued series of intramolecular reactions result in a double-bonded C–N group, which cross-links with other similar mucin compounds [13]. This chemical reaction results is an antigen substance detected by IgE antibodies, which, in turn, triggers an inflammatory response—the root cause of airway inflammation [13]. Mucin compounds are commonly found in the eyes and esophageal lining, both common locations for irritation and dry skin incidence.

Respiratory health effects of (indoor) VOCs have been the subject of research scrutiny. Several review articles and meta-analyses have already been published [14,15,16]. However, there remains limitations to these studies. First, the relative inconsistency in the conclusion of these studies. In a review by Daisey et al., published in 2003, several of the relevant studies included found positive correlations between asthma and VOCs, which prompted them to conclude that schools should take measures to improve ventilation to reduce VOC concentrations in classrooms [17]. This is in contrast to a meta-analysis conducted by Canova et al., published in 2013. They concluded that there was inadequate evidence of the association between asthma and VOCs [14]. A review by Paciênciaa et al. concluded that high concentrations of VOCs are found indoors, but they did not observe any association between VOCs and asthma. Another review by Nurmatov et al., published in 2015, concluded inadequate evidence of the association between VOCs and asthma [15,16]. Second, most of these reviews or meta-analyses included a limited number of studies and lacked empirical data in their conclusions. Third, most of these studies were published before 2016. A few studies have also been published in recent years focusing on the association between VOCs and asthma. In Rufo et al., Daisey et al., Canova et al. and Paciênciaa et al. the number of studies incorporated in their meta-analyses were six, four, 25 and 40, respectively, and a review by Nurmatov et al. incorporated 50+ studies [14,15,16,17,18]. Finally, none of the mentioned meta-analyses or reviews found solid evidence to infer that there was an association between indoor air pollutants and asthma [14,15,16,17] due to tremendous inconsistency in the results of the studies included in these meta-analyses.

Leveraging the most recent literature (up to 2020) and methods of standardization of the effects size, this meta-analysis aims to address the above research gaps. Specifically, this meta-analysis increases the sample size to 49 relevant studies, which includes empirical data published since 2011 included in Paciênciaa et al.’s review [16], which was published in 2016. Thus, this meta-analysis offers an up-to-date account of the association between indoor VOCs and asthma. This includes all studies which were relevant to our meta-analysis. The remainder of this paper includes materials and methods used in this study, results of the meta-analysis, and a discussion on the findings of our paper vis-à-vis the available literature.

## 2. Materials and Methods

This study conducted an extensive literature review using the PubMed research database through EndNote software published in English. We searched PubMed, a central database for articles, with the following key words in the abstract or title: (“volatile organic compound” OR “VOC”) AND (“building material” OR “indoor air quality” OR “sick building syndrome”). We used the following criteria to select studies that were included in the meta-analysis: studies which investigated the role of airborne exposure to VOCs in indoor environments in asthma morbidity and/or related pulmonary diseases; studies that included empirical data and results in the form of coefficients, including odds ratios, relative risk and correlation.

Review articles and/or meta-analyses, and articles that reported inconclusive results or that did not measure the levels of VOC directly were excluded. Articles that reported the use of VOCs as a biomarker or as exhaled biomarkers were also excluded, as the exhalation of VOCs causing asthma development was not a focus of this study. After preliminary screening, a total of 246 articles were evaluated for inclusion. However, after reviewing these articles, only 49 studies met the inclusion criteria and were included in the meta-analysis. Only some variables were also noted, including types and concentration of causal (or exposure) variables, sample size, country where the study was conducted, age and gender composition of the sample. This is because these variables are shown to modify and/or mediate the effects of VOCs exposure. For example, the sample size affects the Type 2 error. Likewise, the effect of environmental pollutants, including VOCs, varies by age, gender and region. Moreover, these variables were comparable and available for most studies included in the analysis. Other socio-demographic variables, such as income and occupation, were not included. Even though these variables can confound the effects of VOCs on respiratory diseases, such variables were either not available or not comparable across studies.

Empirical results were found in each study, such as odds ratios (OR) and correlation coefficients (CC). There were variations in different methods of analyzing data and reporting, such as logistic regression, linear regression and correlation. Thus, we standardized these results by computing the standardized effect size using the methodology presented in Borenstein et al. [19]. The effect size that indicates the strength of association between two variables is grouped under three categories: small = 0.1 to 0.3, medium = 0.3 to 0.5 and large = 0.5 to 1. All analyses were conducted in STATA 14.2 (STATA, College Station, TX, USA) [20].

## 3. Results

A total of 49 studies were included in the meta-analysis. The study characteristics, including VOC types and health outcome(s), of these studies are summarized in Table 1. 37 (87.7%) of the 49 studies were epidemiological, and the remaining 12 were clinical and cross-sectional studies. 15 studies included smokers, and the average number of smokers in these studies was 16.3%. 46 of 49 studies reported ORs, and three of them reported a coefficient of correlations between VOC and pulmonary diseases. 42 of the 49 studies provided sample composition by gender and 49.3% of the sample represented male population (95% confidence interval (CI) 44.8% to 53.9%). Out of the 49 studies, adult subjects without any criteria for a given disease were most frequently included in these studies (*n* = 23 studies).

A total of 13 air pollutants were included across these studies which were further grouped into four groups: total VOCs (TVOC), multiple VOCs (MVOC), specific VOCs (such as formaldehyde, benzene and toluene), criteria pollutants (such as CO, PM_2.5_) and paint exposure. 19 (or 38.7%) of the 49 studies assessed TVOCs and 18 (36.7%) of the 49 included specific VOCs. Different health outcomes were grouped into four categories: asthma (23/49), wheezing (10/49), throat irritation (8/49) and others (8/49). About half of these studies focused on the pediatric population and the remaining studies included all age groups or the 18+ population, as summarized in Table 2.

The study specific effect sizes of the association of VOCs and selected diseases or symptoms are presented in Figure 1. The average effects size, irrespective of air pollution and health outcome types, across 49 studies was 0.37, a medium effect size (95% confidence interval (CI) = 0.29–0.44). This effect size varied across countries (Table 3). The highest effects of VOCs on any of the four outcomes was observed in Iran, followed by Korea, Germany, Sweden, The Netherlands and France. The lowest effect of VOCs was observed in Argentina, followed by Australia, South Africa and Canada. The effect size in the US was 0.36, which is close to the average effect size of all studies.

The effect size also varied by disease/symptom (Figure 2A) and VOC types (Figure 2B) as well as by age (Table 4). It is interesting to note that the effects of VOCs was more than two times higher than the effects of criteria pollutants (Figure 2B). Specific VOCs, such as formaldehyde and benzene, had the highest effect size. Although a limited number of studies have examined the association between indoor VOCs and respiratory health in the elderly population, the highest effect size was observed in the elderly population. It is interesting to note that the effect of VOCs in the adult population was 1.6 times higher than in the pediatric population (effect size 0.3 in the pediatric population versus 0.5 in the adult population).

## 4. Discussion

The results of this study show that VOC exposure has a medium-level association with pulmonary diseases, suggesting that indoor VOC exposure is a moderate risk factor for the onset of pulmonary diseases, including asthma and its associated symptoms, such as wheezing and throat irritation. The effect size varied by country, age and disease/symptom type. Among the confounding factors, we did not find a correlation between smoking and the development of asthma, as the findings of the studies included in the meta-analysis were inconsistent: studies with no smokers had higher effect sizes vis-à-vis those with smokers. Similarly, no correlation between ambient air pollution and the development of respiratory diseases was found. Despite The Netherlands ($\bar{x}$ = 12.1 µg/m^3^; 95% CI = 11.5–12.6 µg/m^3^), Germany ($\bar{x}$ = 11.7 µg/m^3^; 95% CI = 11.7–12.2 µg/m^3^) and Sweden ($\bar{x}$ = 5.9 µg/m^3^; 95% CI = 5.8–6.5 µg/m^3^) having low mean air pollution concentrations of PM_2.5_, studies from these countries exhibited higher effect sizes for asthma development or exacerbation [70]. The studies from China ($\bar{x}$ = 49.2 µg/m^3^; 95% CI = 49.7–53.8 µg/m^3^) with high mean air pollution concentration, showed a small effect. Despite these unexpected anomalies, the findings of this study have public health implications.

VOCs are commonly found in many consumer products and ubiquitously found indoors in HICs [10]. According to the United States Environmental Protection Agency (USEPA), in the United States, the concentration of VOCs is 2 to 5 times higher inside homes than outside [10]. Until recently, our major focus has been on airborne particulate matter (PM). However, the biological mechanisms and management strategies of mitigating exposure to PM are different from that of VOCs. While PM exposure induces oxidative stress [71], VOC exposure induces high chemical reactivity with the amine-rich epithelium lining and mucosa membrane through C–O polar bonding mechanism and subsequent inflammation [13]. Thus, VOC mitigation strategies need to factor in the chemical composition of VOCs. For example, high efficiency air purifiers with an activated carbon filter can reduce VOC concentration [72]. However, it will not eliminate VOC sources, as consumer products continually emit VOCs for months to years. Thus, such filters need to be operational continuously to reduce VOCs. Additional strategies are warranted to target VOC sources by engaging multiple stakeholders in bringing awareness about the danger of VOC exposure, publicly accessible information on consumer products that have VOCs and stringent regulations of consumer products that have VOCs.

The findings of this study must be interpreted in the context of its limitations. First, a limited number of studies were included in the analysis because many studies in the searched literature did not include empirical data on VOCs and respiratory health outcomes. Second, most of these studies were conducted in HICs, with the exception of a few studies in middle-income countries (MICs). Thus, the findings of this study may not be generalizable to the populations of low-income countries (LICs) and MICs, because of striking differences in the levels and types of indoor air pollution in LICs and MICs in comparison to those found in HICs. Moreover, the building design and duration of time people spend indoors also varies across HICs and LICs. Buildings in HICs are sealed for energy conservation and there is less exchange between indoor and outdoor air; consequently, the indoor air pollution concentration builds up over time. However, in LICs there is more indoor and outdoor air exchange because people leave their windows and doors open for ventilation. While it can reduce indoor VOCs from the consumer products, it may increase other air pollutants from outdoor sources if they are found in high concentration outdoors, such as ambient PM found in China and India [73]. Moreover, people spend more time indoors in HICs, and hence are more prone to chronic exposure from indoor air pollution rather than outdoor air pollution. Therefore, people in HICs may bear a relatively higher burden of pulmonary diseases associated with indoor VOCs and other pollutants in comparison to those in LICs and MICs. This was also observed in our study, as the effect size of the association between VOCs and pulmonary health was lower in China with relatively high levels of ambient PM_2.5_ as compared to HICs with relatively low levels of ambient PM_2.5_. Third, the sample of studies included from some of the countries was small. For example, only one study each from Argentina and South Africa met the inclusion criteria to be included in the analysis. These countries reported the lowest effects of VOCs on pulmonary diseases. Low exposure to other air pollutants found in these countries can result in downward bias in the effects of VOCs. Moreover, study setting (rural versus urban), sample demographics (age and gender), consumer product choices, local legislations for controlling indoor and outdoor air pollutants and preexisting comorbidities also varied by studies included in our analyses. Therefore, the country-specific effect sizes of VOCs must be interpreted with caution and these factors must be taken into consideration while interpreting the findings reported in this paper. Finally, we have a limited number of studies on all pulmonary diseases and diseases associated with long-term VOC exposure. There were only two studies that examined lung cancer and VOC exposure. Likewise, there was only one study that focused on sleep breathing disorder (SBD), which is shown to mediate the effect of indoor air pollution in pulmonary diseases. Although we found convincing evidence of a medium-sized association between VOCs and pulmonary diseases, further research is warranted to address the above limitations.

## 5. Conclusions

Based on up to date literature, this paper documents a medium-sized association between VOCs and pulmonary diseases. However, the strength of this association varies by country, age and disease/symptom type. The findings of this paper must be interpreted with caution because most studies included in the meta-analysis were conducted in HICs, and all potential pulmonary diseases were not included. Nonetheless, VOC exposure was associated with asthma and related symptoms, including wheezing and throat irritation. Given VOCs are ubiquitously found indoors, proactive strategies are needed to address this emerging indoor air pollutant to mitigate and manage its associated disease burden.

## Figures and Tables

**Figure 1 ijerph-18-01578-f001:**
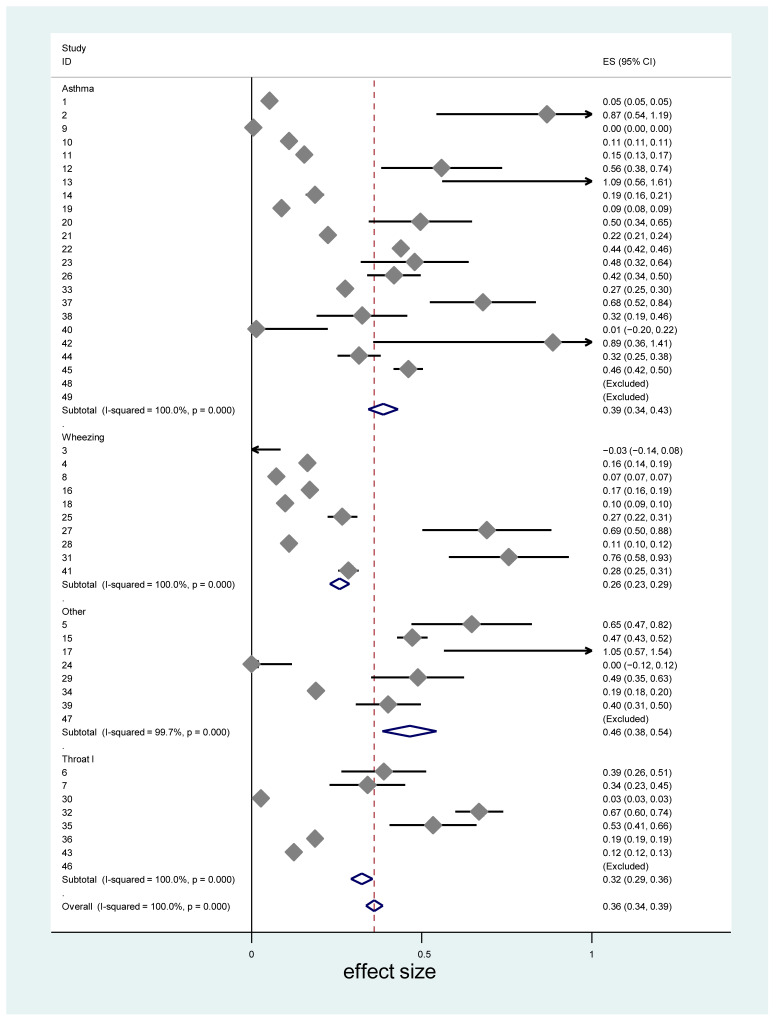
Study specific effect size of the association between volatile organic compound (VOC) exposure and four different health outcomes: development of asthma, wheezing, throat irritation and others (which included respiratory symptoms and diseases). The red dash line represents the average effect size.

**Figure 2 ijerph-18-01578-f002:**
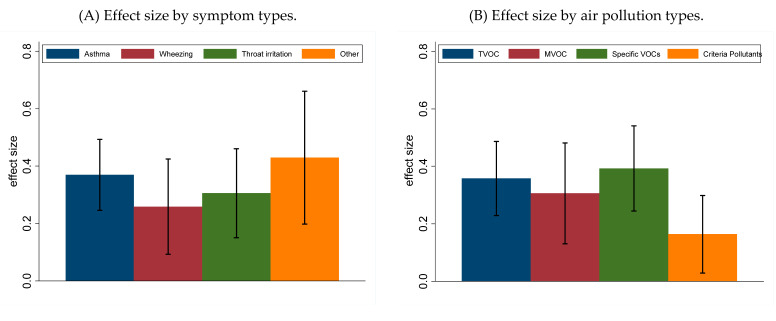
Effect size distribution by disease/symptom types (**A**) and air pollution types (**B**).

**Table 1 ijerph-18-01578-t001:** List of studies included in the meta-analysis. (TI = throat irritation; SBD = sleep breathing disorder; US = United States; UK = United Kingdom; COPD = chronic obstructive pulmonary diseases; all ages: no specific age break up was available).

ID *	Ref. ID	Country	Study Year	Outcome Variable	Sample Size	Age (Year)	Effect Size
1	[21]	France	2011	Asthma	1012	All ages	0.05
2	[22]	France	2019	Asthma	109	All ages	0.87
9	[23]	Australia	2002	Asthma	192	1–18	0.00
10	[24]	US	2003	Asthma	88	1–18	0.11
11	[25]	Australia	1999	Asthma	148	1–18	0.15
12	[26]	France	2013	Asthma	95	1–18	0.56
13	[27]	France	2010	Asthma	114	1–18	1.09
14	[28]	Sweden	2010	Asthma	400	1–18	0.19
19	[29]	France	2012	Asthma	6590	1–18	0.09
20	[30]	France	2013	Asthma	897	18–60	0.50
21	[31]	US	2007	Asthma	550	18–60	0.22
22	[32]	China	2013	Asthma	268	All ages	0.44
23	[33]	Finland	2008	Asthma	137	1–18	0.48
26	[34]	US	2010	Asthma	1480	All ages	0.42
33	[35]	France	2013	Asthma	4209	1–18	0.27
37	[36]	Japan	2004	Asthma	317	18–60	0.68
38	[37]	US	2003	Asthma	24	1–18	0.32
40	[38]	Argentina	2009	Asthma	1183	1–18	0.01
42	[39]	Japan	2011	Asthma	393	60+	0.89
44	[40]	Sweden	2007	Asthma	1014	1–18	0.32
45	[41]	Europe	2020	Asthma	5175	1–18	0.46
48	[42]	China	2020	Asthma	120	18–60	0.16
49	[43]	The Netherlands	2017	Asthma	23	All ages	0.21
**Asthma** = effect size ~0.37; 95% CI = 0.25–0.49; *n* = 23							
3	[44]	Australia	1995	Wheezing	863	1–18	−0.03
8	[45]	Canada	2018	Wheezing	2900	60+	0.07
4	[46]	UK	2008	Wheezing	7162	1–18	0.16
16	[47]	UK	2006	Wheezing	200	1–18	0.17
18	[48]	China	2008	Wheezing	1993	1–18	0.10
25	[49]	United Arab Emirates	2012	Wheezing	1590	All ages	0.27
27	[50]	Iran	2019	Wheezing	456	18–60	0.69
28	[51]	South Africa	2017	Wheezing	1065	1–18	0.11
31	[52]	Germany	2014	Wheezing	465	1–18	0.76
41	[53]	Portugal	2008	Wheezing	1607	1–18	0.28
**Wheezing** = effect size ~0.26; 95% CI = 0.10–0.42; *n* = 10							
6	[54]	Malaysia	2017	TI	462	1–18	0.39
7	[55]	Japan	2010	TI	120	60+	0.34
30	[56]	China	2015	TI	417	18–60	0.03
32	[57]	Korea	2014	TI	159	18–60	0.67
35	[58]	Japan	2012	TI	3950	18–60	0.53
36	[59]	Japan	2009	TI	343	All ages	0.19
43	[60]	Japan	2018	TI	107	All ages	0.12
46	[61]	US	1995	TI	4	1–18	0.17
**Throat irritation** = effect size ~0.31; 95% CI = 0.15–0.46; *n* = 8							
15	[62]	UK	2003	Rhinitis	626	18–60	0.47
24	[63]	Japan	2012	Rhinitis	609	All ages	0.00
29	[64]	Portugal	2016	Rhinitis	143	60+	0.49
5	[65]	US	2003	Bronchitis	186	1–18	0.65
17	[66]	Sweden	1995	SBD	88	18–60	1.05
39	[67]	Germany	2000	COPD	649	1–18	0.40
34	[68]	Canada	2014	Lung cancer	445	Others	0.19
47	[69]	China	2020	Lung cancer	0	Others	0.18
**Others** = effect size ~0.43; 95% CI = 0.20–0.66; *n* = 8							

***** represents ID shown in Figure 1.

**Table 2 ijerph-18-01578-t002:** Sample age by reported health outcomes of the studies included in the meta-analysis.

Age Group (Year)	Asthma	Wheezing	Throat Irritation	Other	Total
1–18	13	7	2	2	24
18–60	4	1	3	2	10
60+	1	1	1	1	4
All ages (non-specific)	5	1	2	1	9
Others	0	0	0	2	2
Total	23	10	8	8	49

**Table 3 ijerph-18-01578-t003:** Effect size (Cohen’s d) by country.

Country Name	Effect Size (95% CI; *n*)
Australia	0.04 (−0.07–0.15; 3)
Japan	0.39 (0.16–0.63; 7)
France	0.49 (0.20–0.78; 7)
Canada	0.13 (0.02–0.25; 2)
Sweden	0.52 (−0.01–1.05; 3)
United Kingdom	0.27 (0.07–0.47; 3)
United States	0.36 (0.21–0.50; 6)
Malaysia	0.39 (NA, 1)
China	0.28 (0.10–0.45; 5)
Finland	0.48 (NA, 1)
United Arab Emirates	0.27 (NA, 1)
Iran	0.69 (NA, 1)
South Africa	0.11 (NA, 1)
Portugal	0.39 (0.19–0.59; 2)
Germany	0.58 (0.23–0.93; 2)
Korea	0.67 (NA, 1)
Europe	0.46 (NA, 1)
Argentina	0.01 (NA, 1)
The Netherlands	0.52 (NA, 1)
Total	0.37 (0.29–0.44; 49)

NA = 95% confident interval was not computed, because of an inadequate number of studies.

**Table 4 ijerph-18-01578-t004:** Disease/symptom specific effects size by age groups.

Disease/Symptoms	Age Group			All Ages (Non-Specific)
	1–18	18–60	60+	
Asthma	0.31 (0.15–0.47; 13)	0.39 (0.15–0.63; 4)	0.89 (NA, 1)	0.40 (0.13–0.67; 5)
Wheezing	0.22 (0.03–0.41; 7)	0.69 (NA, 1)	0.07 (NA, 1)	0.27 (NA, 1)
Throat Irritation	0.28 (0.07–0.49; 2)	0.41 (0.03–0.79; 3)	0.34 (NA, 1)	0.15 (0.09–0.22; 2)
Other	0.52 (0.28–0.77; 2)	0.76 (0.19–1.33; 2)	0.49 (NA, 1)	-
Total	0.30 (0.19–0.41; 24)	0.50 (0.31–0.69; 10)	0.45 (0.11–0.78; 4)	0.29 (0.11–0.46; 9)

## Data Availability

All data used in the paper are publically available. We may supply the complied data that we gathered from public sources upon a reasonable request to the corresponding author.

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
