# Peer review of "Pulmonary Health Effects of Indoor Volatile Organic Compounds—A Meta-Analysis"

_ijerph, 2021, doi:10.3390/ijerph18041578_

Round 1

Reviewer 1 Report

This article describes the pulmonary health effects of indoor volatile organic compounds by using a meta-analysis of literature searched using the PubMed database.  The authors make a compelling assertion that this topic warrants attention due to the worldwide morbidity, mortality and rising healthcare costs associated with chronic pulmonary diseases.  The authors aim to add to the literature by examining pulmonary health effects of indoor volatile organic compounds (VOCs) as they report previous meta-analyses produced inconsistent findings and were published before 2016.

In lines 123-125, authors describe how the important variables in the 49 studies that met the exclusion criteria are noted.  Authors mention that they noted the important variables, including “types and concentration of causal (or exposure) variables, sample size, country where the study was conducted, age and gender composition of the sample”.  It may be helpful here for authors to provide an explanation as to why these specific variables were chosen.  Where there any other variables considered?  Some other explanatory variables that come to mind include income, education, occupation, and geographic location.

In lines 141-142, authors mention that healthy adults were often included in the studies (n=23).  What constituted being a ‘healthy adult’?  Was this a self-reported measure?

In line 155-159, the findings are intriguing.  Authors discuss how “the lowest effect of VOCs was observed in Argentina followed by Australia, South Africa and Canada”.  Perhaps the authors can expand on these findings in the discussion section.  Is there literature that gives insight into what factors may contribute to lower effect of VOCs in these particular countries?

Overall, this is an interesting, novel, and insightful study.  The study can add to the literature as authors state that previous studies on this topic have produced inconsistent findings.  Therefore, attending to clarifying questions, including about the variables explored in this study, may help to improve the paper.

Author Response

AR: We greatly appreciate your time for reviewing our manuscript and providing us with your constructive suggestions and comments.

In lines 123-125, authors describe how the important variables in the 49 studies that met the exclusion criteria are noted.  Authors mention that they noted the important variables, including “types and concentration of causal (or exposure) variables, sample size, country where the study was conducted, age and gender composition of the sample”.  It may be helpful here for authors to provide an explanation as to why these specific variables were chosen.  Where there any other variables considered?  Some other explanatory variables that come to mind include income, education, occupation, and geographic location.

AR: Some of the variables included in the analysis were available for most studies and these variables can affect the robustness of a study as well as can confound the effects of VOCs no pulmonary disease. For example, the sample size affects Type 2 error, thus it is was important to include it in the meta-analysis. Likewise, the effect of environmental pollutants, including VOCs, can vary by age, gender and region. That is why these variables were included in the analysis. You are correct that other socio-demographic variables that you mentioned can also influence the effects of VOCs on respiratory diseases, but these were not comparable across studies and not available for most studies. We expanded on these two points, see lines 126-133 in the revised manuscript.

In lines 141-142, authors mention that healthy adults were often included in the studies (n=23).  What constituted being a ‘healthy adult’?  Was this a self-reported measure?

AR: Health adults refers to adult subjects in age 18 or older without any criteria for a given disease diagnosis. See lines 149-150 

In line 155-159, the findings are intriguing.  Authors discuss how “the lowest effect of VOCs was observed in Argentina followed by Australia, South Africa and Canada”.  Perhaps the authors can expand on these findings in the discussion section.  Is there literature that gives insight into what factors may contribute to lower effect of VOCs in these particular countries?

AR: Given the lack of literature, it is hard to explain underlying reasons for this observation. But we have included two potential explanations: small samples size and low exposure to other air pollutants that can reduce synergistic effects of VOCs. See lines for 244-253 in the revised manuscript.

Overall, this is an interesting, novel, and insightful study.  The study can add to the literature as authors state that previous studies on this topic have produced inconsistent findings.  Therefore, attending to clarifying questions, including about the variables explored in this study, may help to improve the paper.

AR: We greatly appreciate your feedback and comments.

Reviewer 2 Report

This is an up to date meta-analysis of pulmonary health effects of indoor VOC exposures. The article is well written except for some minor edits required as outlined at the bottom of this section and a few issues, including

1) The presentation of Figure 2 before Fig 1 in the text; if this order is maintained they should be switched around.

2) the need to clarify what is meant when the authors state they omitted studies that reported "inconclusive" results (line 117). It should be very clearly explained why any studies were left out.

3) Fig. 1: Define the meaning of the dashed red line in Fig 1. Define ES (effect size) at the top of that chart. better explain the column  % weight in the legend.

Minor edits:

Sentence 41-42 seems to be missing words.

Line 75: Mucin compounds "are" commonly missing...

Line 77: ..have been "the" subject of..

Line 164: Although "a" limited...

Line 165: "The association..."the " elderly..

Line 166: "The" highest effect size..

Line 167: "the" adult population..

Fig 2 is split over 2 pages

Line 200: "bringing" is misspelled

Lines 207-208: sentence should be corrected

Line 225: "lung cancer" should have no hyphen

Author Response

Reviewer 2

Comments and Suggestions for Authors

This is an up to date meta-analysis of pulmonary health effects of indoor VOC exposures. The article is well written except for some minor edits required as outlined at the bottom of this section and a few issues, including

AR: Thank you for your comments.

1) The presentation of Figure 2 before Fig 1 in the text; if this order is maintained they should be switched around.

AR: Figure placement is rearranged as suggested.

2) the need to clarify what is meant when the authors state they omitted studies that reported "inconclusive" results (line 117). It should be very clearly explained why any studies were left out.

AR: This refers to non-significant findings of the results.

3) Fig. 1: Define the meaning of the dashed red line in Fig 1. Define ES (effect size) at the top of that chart. better explain the column  % weight in the legend.

AR: The red dash lined represented the overall average effect size of all studies included in the analysis. Since all studies were weighted equally, % weight is now removed from the figure.

Minor edits:

Sentence 41-42 seems to be missing words.

AR: Now it reads, Multiple stakeholders must be engaged in strategies to, …

Line 75: Mucin compounds "are" commonly missing...

AR: Edited as suggested.

Line 77: ..have been "the" subject of..

AR: It is changed to make it more readable. Now, it reads as “have the subject of research …”

Line 164: Although "a" limited...

AR: Corrected as suggested.

Line 165: "The association..."the " elderly..

AR: corrected as suggested.

Line 166: "The" highest effect size..

AR: Edited as suggested.

Line 167: "the" adult population..

AR: Corrected as suggested.

Fig 2 is split over 2 pages

AR: Fixed!

Line 200: "bringing" is misspelled

AR: corrected!

Lines 207-208: sentence should be corrected

AR: Edited as suggested.

Line 225: "lung cancer" should have no hyphen

AR: Edited as suggested.